# Implementing the IMPALA continuous monitoring system for paediatric critical care in Malawi: A mixed methods study of barriers and facilitators

Daniel Mwale [1,2,3,4*], Christopher Pell[3,4,5], Jobiba Chinkhumba[1], Happiness Mandala[1], Jessica Chikwana[6], Josephine Langton [1], Alice Likumbo[7], Michael Boele van Hensbroek[2,3,8], IMPALA Study Team†, Job Calis[1,3,9], Wendy Janssens[3,10], Lucinda Manda-Taylor[1,7]

1 Kamuzu University of Health Sciences, Private Bag 360, Chichiri, Blantyre, Malawi, 2 Amsterdam Centre for Global Child Health, Emma Children's Hospital, Amsterdam UMC, Meibergdreef, the Netherlands, 3 Amsterdam Institute for Global Health and Development, Amsterdam, The Netherlands, 4 Amsterdam UMC, location University of Amsterdam, Department of Global Health, Amsterdam, the Netherlands, 5 Amsterdam Public Health Research Institute, Amsterdam, the Netherlands, 6 Zomba Central Hospital, 7 Training Research Unit of Excellence, Malawi, 8 Department of Paediatric Infectious Diseases, Emma Children's Hospital, Amsterdam UMC, Meibergdreef, the Netherlands, 9 Department of Paediatric Intensive Care, Emma Children's Hospital, Amsterdam UMC, Meibergdreef, the Netherlands, 10 Dept of Economics, Vrije Universiteit Amsterdam, the Netherlands

† See Annexe[1]
* dmwale@kuhes.ac.mw

## Abstract

### Introduction

Continuous monitoring of critically ill children is essential for the timely identification of deteriorating vital signs. However, monitoring is often intermittent in low-resource settings, affecting the quality of care. This study assessed the implementation barriers and facilitators of a locally adapted, robust, low-cost continuous monitoring system (IMPALA) in Malawi.

### Methods

A mixed-method implementation study of the IMPALA system in the paediatric High-dependency unit of a tertiary hospital from November 2022 to October 2023. Data were collected through over 300 hours of observations, in-depth interviews with 14 healthcare providers and nine caregivers of admitted children, and questionnaire-based surveys from 24 healthcare providers and 72 caregivers. Qualitative data were analysed thematically using inductive and deductive approaches. Descriptive statistics (frequencies, percentages, means, and standard deviations) were calculated for categorical and continuous variables.

### Results

Healthcare providers and caregivers indicated that the IMPALA monitors improved care by providing the ability to measure reliably multiple vital signs, with long-lasting

**Data availability statement:** The minimal anonymized dataset necessary to replicate the study findings has been uploaded to a stable, public repository and the accession numbers or DOIs necessary to access these data is as follows: https://www.openicpsr.org/openicpsr/project/236881/version/V1/view. The project is under openICPSR Project openicpsr-236881.

**Funding:** The European and Developing Countries Clinical Trials Partnership 2 programme supported by the European Union (Grant number RIA20201-3294 – IMPALA) provided all the funding or sources of support. There was no additional external funding received for this study. The grant was awarded to JC. The funders had no role in study design, data collection and analysis, decision to publish, or preparation of the manuscript. edctp.org.

**Competing interests:** I have read the journal's policy and the authors of this manuscript declare that they have no known competing financial interests or personal relationships that could have appeared to influence the work reported in this paper. The authors declare the following financial interests/personal relationships, which may be considered as potential competing interests; financial support, administrative support, equipment, supplies, travel, and writing assistance were provided by Kamuzu University of Health Sciences.

(4 hours) backup power and alarm provisions. Healthcare providers reported spending less time on child monitoring after the introduction of IMPALA (1.8 hours per day pre-IMPALA (95% CI: 1.19–2.48) compared to 3.3 hours post-IMPALA (95% CI: 2.36–4.23; p < 0.00). Still, they recognised alarm fatigue, limitations in knowledge of the technology, and staff shortages as barriers to the use of IMPALA. Some caregivers expressed concerns about the reliability of the monitoring system.

## Conclusion

The continuous monitoring device was well-received overall by healthcare providers and caregivers. It was perceived to save time and improve the quality of care. Opportunities to further enhance engagement with the device include strengthening caregivers' knowledge and involvement to address their mistrust or misconceptions about the device, minimising false alarms, and providing ongoing training to healthcare providers so that new, existing, and rotating staff know how to engage with the device.

## Introduction

In Malawi and other low- and middle-income countries (LMICs), hospital care for critically ill children is often sub-optimal [1]. Critical care services are often not accessible, and where care in high-dependency units (HDUs) or paediatric intensive care units (PICUs) is available, the health outcomes of admitted children are often worse than in high-income settings [2,3]. Three studies conducted in various tertiary hospitals with HDUs in Malawi reported paediatric hospital mortality rates ranging from 2.2% to 4.4%, which aligns with the average paediatric hospital mortality rate of 4.1% reported in a recent meta-analysis of studies conducted in LMICs [4–7]. A major cause of in-hospital mortality is related to the late detection of deteriorating vital signs [8]. Monitoring the vital signs of admitted children is crucial in providing high-quality critical care, enabling healthcare staff to identify deterioration early and respond promptly [9]. When followed by prompt and appropriate action, close monitoring can reduce mortality, morbidity, hospital length of stay and healthcare costs [10]. Various factors can influence vital sign monitoring, including the availability and usability of monitoring devices, clinical shift duration [11], ward staff training levels, the number of staff [12], standard operating procedures (SOPs), and workload.

However, devices that continuously monitor vital signs are not readily available or contextually suitable for many LMIC settings. This is often due to the prohibitive costs of purchasing and maintaining complex devices, which are a poor fit for the critical care infrastructure in LMICs, and the limited training of healthcare staff [11]. The existing monitoring systems are not designed to cope with power surges or electricity failures, are too complex, and are unsuitable for overburdened staff in settings with low staff-to-patient ratios. So, healthcare providers in low-resource settings usually

employ manual techniques or rely on spot-checking equipment such as thermometers and pulse oximeters to capture intermittent vital signs in hospitalised paediatric children [12,13].

To address these challenges, a multidisciplinary team of designers, researchers and intensive care paediatricians has been developing and testing a continuous monitoring device called IMPALA. IMPALA stands for "An Innovative Monitoring system for PAediatrics in Low-resource settings: an Aid to save lives." The device has been designed to be robust, easy to use, and affordable. IMPALA was developed using a user-centred design approach, incorporating the needs of nurses, doctors and clinical officers, such as visual alerts, adjustable alarm thresholds and simplified language. The device monitors vital signs, including heart rate, respiratory rate, oxygen saturation and blood pressure, in critically ill children aged between 28 days to 5 years.

Despite its potential, implementing novel health technologies in LMICs often faces challenges, and end-user utilisation or the anticipated impact on health outcomes cannot be taken for granted especially in LMICs, where health facilities are often underfunded and inefficient [14]. There remains limited evidence on facilitators of and barriers to the successful implementation of new technologies in low-resource hospital settings. To address this gap, a pilot study of the locally adapted IMPALA 2.0 (hereafter referred to as IMPALA) monitoring system in the paediatric HDU of a tertiary referral hospital in Malawi was conducted before and after implementation to examine how the IMPALA monitors were perceived and used and what factors affected their use during implementation. This article draws on data collected through in-depth interviews, questionnaire-based surveys, and structured observations to understand the perceptions and behaviours of healthcare providers and caregivers of children admitted to the HDU and to identify barriers to and enablers of continued use of the monitoring system in an LMIC setting. The aim is to inform the further improvement of the IMPALA system (IMPALA 3.0), enhance its uptake and use in Malawi, and describe lessons for effectively implementing such systems in low-resource settings elsewhere.

## Methods and setting

### Study design

This implementation study used mixed methods to assess the use of the IMPALA continuous monitoring system during an observational study and the barriers and facilitators to its implementation. It included observations of critically ill children and their caregivers and healthcare providers, in-depth interviews and quantitative surveys with caregivers and healthcare workers, and user implementation diaries of the study nurses and clinical staff. A mixed-methods design was chosen for the following reasons: observations and interviews were conducted to mitigate any potential desirability bias that might arise from the observations. At the same time, surveys were conducted to examine and quantify the responses from the wider set of respondents. The triangulation of data sources enhanced methodological rigour and strengthened the validity of the findings.

This study builds on the insights from the qualitative research conducted before implementing the IMPALA continuous monitoring system in the same setting to understand monitoring practices in the status quo [1]. The study is registered at https://www.isrctn.com/ISRCTN71392921.

### The intervention

The IMPALA monitor was developed as an affordable, durable and user-friendly paediatric continuous monitoring system for low-resource settings. Unlike conventional systems, IMPALA features a novel, context-specific design with adjustable alarms, configurable settings for end-users, a scroll-based menu, and a simplified interface. Its power-independent operation, enabled by up to four hours of battery backup, makes it uniquely suited for environments where electricity and infrastructure are often unreliable. It was implemented as part of a one-year observational study (from 8th July 2022–30th June 2023) that recruited a cohort of 774 children (aged between 28 days and 60 months) admitted to HDUs in Zomba Central Hospital, Zomba, and Queen Elizabeth Central Hospital, Blantyre (Fig 1).

Clinical staff could see children's vital signs and adjust their clinical decision-making behaviour. The hospital staff (nurses, supporting staff, clinicians and doctors) working in Zomba Central's paediatric HDU were trained to use the IMPALA monitoring system and the study. The five-day introduction of the IMPALA study to the hospital staff was divided into four sessions (aimed at nurses, doctors/clinical officers, technicians and supporting staff such as ward attendants and cleaners, respectively). The introduction covered the following areas: acknowledging, adjusting, and pausing alarms; working relationship expectations; and how to intervene in cases of adverse critical illness events. In addition, before the study commenced, a team of five study nurses was specifically trained on the protocol for the observational study, to obtain informed consent and assist hospital staff with connecting and disconnecting monitors. They were also trained on the scrolling the menu, interpreting the vital signs results/readings from the monitors, responding to alarms (distinguishing critical alarms from false alarms and adjusting the alarms), handling basic troubleshooting, carrying out safety usage procedures for the monitor, repeat knowledge on actions to take when alarms occur (using Emergency Triage Assessment and Treatment (ETAT) approach). The training involved practical sessions and demonstrations. Children were connected to the system, and their vital signs (heart rate, respiratory rate, oxygen saturation, and non-invasive blood pressure) were measured to develop algorithms.

## Study setting

The study was conducted in the paediatric HDU of Zomba Central, a tertiary referral hospital in southern Malawi. Zomba is a large rural district in southerns Malawi, with an estimated population of 746,000 [15]. Zomba Central is a government referral hospital for four other regional district hospitals. Hospital policy mandates that all routine care is free to the user [15]. Fig 2 outlines the floor plan of Zomba Central's paediatric ward. This includes an admission room where children are assessed and triaged for admission to the medical or surgical wards. The paediatric HDU has eight beds in two

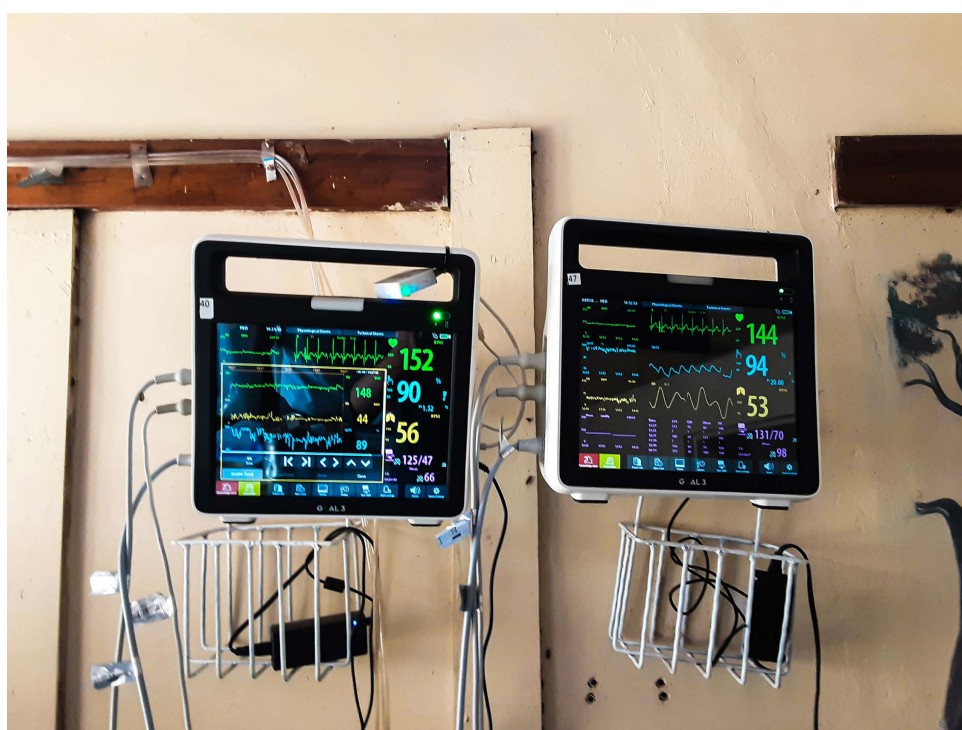

**Fig 1. IMPALA vital signs monitor (prototype 2.0).**

neighbouring rooms in the ward. HDU 1 has a door with three beds and two monitors; HDU 2 has five beds and seven monitors (Fig 2), and an improvised window-level cardboard door. Cardboard room dividers separate the HDUs from the medical ward. During busy times, up to 3 children share a single bed.

Both HDUs admit critically ill children and those needing extra monitoring in addition to the routine six-hour monitoring in the medical ward. The triage system for HDU admission involves assessing airway, breathing pattern, circulation, and convulsions using the Blantyre Coma Scale. Due to the limited number of beds, two or sometimes three children, regardless of their medical conditions or levels of severity, share the same bed. At most, one caregiver per child can be at the admitted child's bedside during the hospitalisation. The HDU admits children below 12 years old, with critically ill neonates admitted to a separate neonatal ward. Around ten children are admitted daily to the HDUs during the rainy season [1]. The most common reasons for admission include malaria, sepsis and pneumonia. In the dry season, daily admissions decline to around six to eight [1].

The paediatric department has two shifts (day and night). Day (8 h) and night (16 h) shifts each have one nurse assigned to the HDUs. A paediatric consultant, an intern doctor, nine nurses and two clinicians are assigned to the paediatric ward on day and night shift rotations. Therefore, five nurses, a paediatric consultant, and two clinicians are assigned to the paediatric department on day shifts. In comparison, night shifts consist of four nurses, one clinician, and an on-call paediatric consultant. The healthcare providers are rotated weekly. Before the start of the intervention, the HDU had two mobile monitors, one pulse oximeter, four oxygen concentrators and one non-functional wall-mounted monitor.

## Conceptual framework

The study was guided by the five domains of the adapted Consolidated Framework for Implementation Research (CFIR) (Fig 3). The CFIR, a meta-framework drawing on various disciplines, offers a comprehensive taxonomy of constructs that affect the implementation of complex interventions. It comprises five major domains: 1) Intervention characteristics (the innovation being implemented), 2) outer setting (external influences), 3) inner setting (the setting in which the innovation is implemented), 4) characteristics of individuals (roles and characteristics of individuals), and 5) process of implementation. This article focuses on intervention characteristics (design quality #1), the outer setting (patient needs and resources #2),

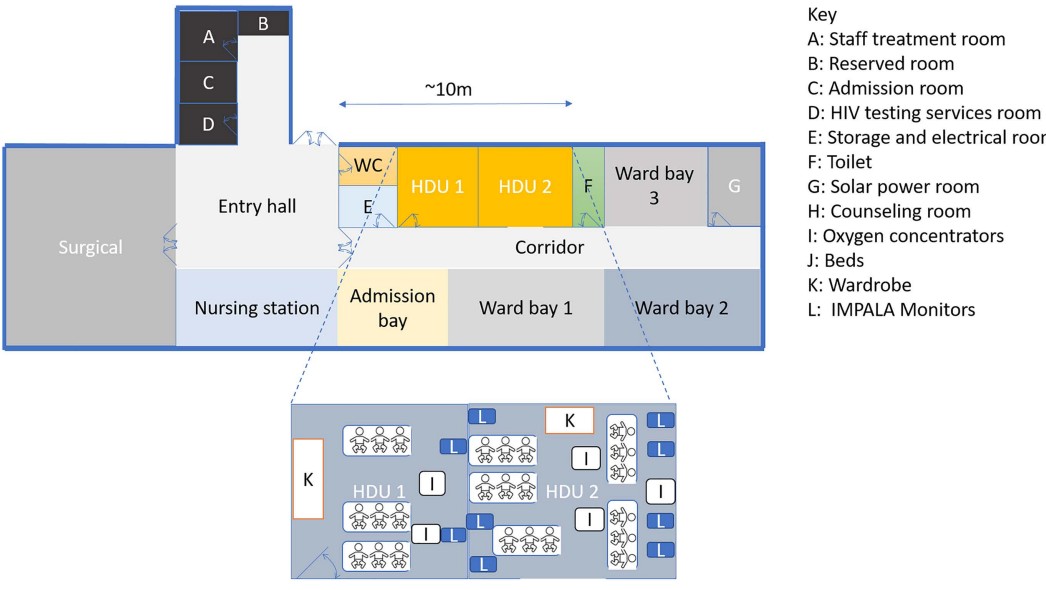

Key
A: Staff treatment room
B: Reserved room
C: Admission room
D: HIV testing services room
E: Storage and electrical room
F: Toilet
G: Solar power room
H: Counseling room
I: Oxygen concentrators
J: Beds
K: Wardrobe
L: IMPALA Monitors

**Fig 2. Floor plan of the paediatric ward.**

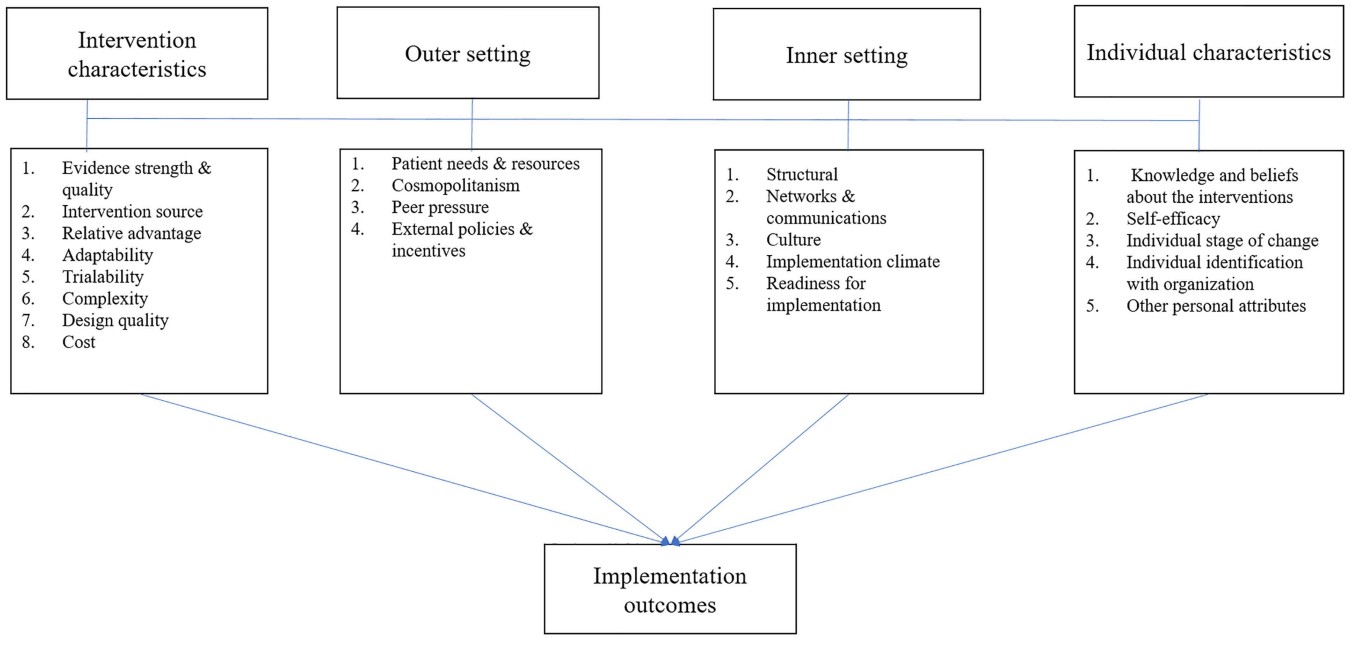

**Fig 3. Adapted Consolidated Framework for Implementation Research (CFIR).**

the inner setting (implementation climate #3) and characteristics of individuals (knowledge and beliefs about the intervention #4).

## Respondent sampling and recruitment

**Observations and in-depth interviews.** In total, 23 observations and in-depth interviews (IDIs) were conducted at Zomba Central Hospital's paediatric HDU, with healthcare providers (n = 14) and caregivers (n = 9) of nine critically ill children (aged 3 months to 5 years old) admitted to the HDU. After introducing the IMPALA monitors, study participants (healthcare providers and caregivers) were conveniently recruited for structured observations and in-depth interviews between November 9, 2022, and October 30, 2023. Healthcare providers of different cadres (nurses, clinical officers and medical doctors) working in HDU were recruited (face-to-face) during their day shifts, representing a mix of professional roles and levels of experience. Selection was based on their involvement in paediatric care, treatment and management. To enhance diversity and representativeness, caregivers of critically ill children with a range of diagnoses and socioeconomic status were also recruited during their admission to HDU. The diversity sample of critically ill children was selected based on the following criteria: age (28 days to 60 months) and categorised as (a) febrile respiratory (e.g., pneumonia, bronchiolitis and asthma), (b) febrile non-respiratory (e.g., malaria and anaemia) or (c) neurological (e.g., head injuries) conditions. These conditions were selected because they are prevalent in paediatric populations in Malawi and have varied typical progressions during admission. Caregivers included anyone accompanying the admitted children during their hospital stay. Caregivers were selected based on being above eighteen years of age and fulfilling a caregiving role. This approach ensured capturing the lived experience of providing care for critically ill children. The recruitment of participants continued until saturation, i.e., when no new or additional information was forthcoming.

**Questionnaire-based survey.** The sampling frame for the questionnaire-based caregiver survey consisted of all caregivers of critically ill children aged three months to five years who were admitted to the HDU between January 1 and June 30, 2023, and consented to participate in the IMPALA clinical observational study. Children under the age of five

had the highest likelihood of presenting with critical illness, and this was the population with the highest mortality. Clinical deterioration was less evident in older age groups, and disease progression develops more rapidly in younger children compared to older ones [16]. Study nurses identified eligible caregivers in collaboration with clinical staff, and additional informed consent was sought to participate in the questionnaire-based survey upon their child's discharge from HDU. Caregivers whose child died in HDU did not participate in the surveys. A maximum of 110 caregivers were expected to be interviewed, dependent on the number of admissions and caregiver consent during the data collection period. The sampling frame for the questionnaire-based healthcare provider survey comprised all healthcare workers (i.e., the complete census of nurses and clinicians) working in the HDU from January to May 2023. We estimated the minimum detectable effect size of the change in hours spent on monitoring children before and after IMPALA for the sample of n = 24 healthcare workers, based on a one-sample mean t-test power calculation with alpha = 0.05, beta = 0.80 and a standard deviation of 1.1025, to be 0.597. Though the sample size was small, it was appropriate for the exploratory, early-stage nature of this study. It was chosen to balance statistical power with feasibility in a resource-constrained setting.

### Data collection

**Observations.**  Two structured observation guides (for children admitted to the HDU and healthcare providers) were used to collect data on various areas of critical care in the HDU, including diagnosis, communication, frequency of monitoring patient vital signs, use of monitors, response to perturbations, treatment rendered, and other tasks. Observations were conducted throughout the day shift, focusing on critically ill children and their caregivers from the point of admission to the HDU until discharge or death. Attending healthcare providers were also observed throughout the entire day shift to capture their interactions, workflows, and use of the monitoring system. A social scientist (DM) conducted the observations with support from the research nurse (AL). The researchers observed each child participant for eight hours and systematically recorded their observations using the observation guides. Structured observations were conducted with ten healthcare providers and nine children and caregivers. Over the course of 40 days, a total of 320 hours of observations were completed.

**In-depth interviews.**  The study team developed interview guides for healthcare providers and caregivers based on the CFIR. These guides covered topics such as experiences during admission, vital signs monitoring processes, and various constructs from the CFIR. The semi-structured guides included open-ended questions that were used flexibly and adapted during data collection. To ensure accessibility, the interview guides and consent forms were translated into the local language by a native Chichewa speaker. Initially, study nurses recruited caregivers of critically ill children admitted to the HDUs who consented to participate in the IMPALA study. Subsequently, DM obtained separate consent for the interviews and conducted them a day after the observation was completed. All interviews with caregivers were conducted in Chichewa, the local language, to facilitate understanding and expression. Some interviews with healthcare providers were conducted in English or Chichewa, depending on the provider's preference. They took place in a private room (such as a nurse's restroom) or a location selected by participants and lasted thirty to forty-five minutes. Healthcare workers were interviewed during lunch breaks or at the end of their shifts to minimise disruption to their daily tasks.

**The questionnaires.**  The self-administered EMPATHIC-65 questionnaire served as the basis for the quantitative survey tool for caregivers [17]. The EMPATHIC-65 questionnaire was initially developed and validated in the Netherlands in a cohort of 3,354 parents of children admitted to eight Dutch PICUs and subsequently adapted and validated in South Africa [18]. The EMPATHIC-65 questionnaire includes sections on general information about PICU patients, parents' satisfaction with the quality of HDU care, and their experiences in the PICU, which were adapted to the study context and translated from English to Chichewa. The questionnaire was expanded to encompass caregiver experiences during admission and vital signs monitoring. The revised questionnaire was extensively piloted among caregivers and further adjusted before data collection. The survey instrument for healthcare providers was specifically designed for this study and included sections on provider background characteristics, training, and current role in the HDU; daily tasks and

time management; and stock outs of medications and medical supplies. The study team administered the tablet-based questionnaire to the healthcare providers in their preferred language, either English or Chichewa. The caregiver survey was always administered in Chichewa.

## Data processing and analysis

**Qualitative data.** The first author, DM – a native Chichewa speaker, conducted the in-depth interviews, which were recorded using a digital audio recorder. Subsequently, the audio-recorded data was translated directly from Chichewa to English by an experienced, independent and bilingual transcriber and native Chichewa speaker. Following transcription, DM meticulously reviewed the transcripts to ensure accuracy by cross-referencing them with the original audio recordings, confirming that the participants' intended meanings were accurately captured. Any information that could identify the study participants was removed from the transcripts. In addition to transcription, DM maintained a field diary, documenting notes on data collection challenges and emerging issues. The audio recordings and complete translated data were securely stored on a password-protected computer. DM listened to the audio recordings and reviewed the transcripts multiple times to gain a comprehensive understanding of the issues raised. Familiarising himself with the entire dataset, DM ensured the data was clean and coherent.

The processes began with importing transcripts from Word documents into NVivo QSR version 14 software (QSR International, Victoria, Australia) for organisation, management, and thematic analysis. Subsequently, DM and LMT collaborated to define terms and concepts extracted from three transcripts, and CP validated them. DM undertook the coding process for all datasets using the agreed-upon coding framework. This collaborative effort guided the development of the codebook utilised for thematic analysis. DM and LMT utilised coding definitions from the Consolidated Framework for Implementation Research (CFIR) during the thematic process, as depicted in S3 Fig. DM systematically coded the textual data, capturing the participants' viewpoints and narratives within the provided framework. This framework offered systematic guidance for assessing the facilitators, barriers, challenges, and opportunities for implementing IMPALA monitors in Malawi.

**Quantitative data.** Frequencies and percentages were calculated for categorical variables; means and associated standard deviations (SDs) were estimated for continuous variables. The analyses include inferential statistics (95% confidence intervals) for all variables, as well as comparative statistics for the before-after difference in hours worked on child monitoring. Analyses were conducted using STATA version 17.

## Ethical considerations

The Kamuzu University of Health Sciences (KUHeS) Research and Ethics Committee granted ethical approval for the research (P.01/22/3552) on 11 March 2022—all the data collected from participants adhered to the Declaration of Helsinki and KUHeS guidelines on Health Research. Permission was obtained from the Zomba Central Hospital. Written informed consent, translated into the local language, was provided to all participants by DM, including caregivers of critically ill children and healthcare providers, for observations, in-depth interviews, and questionnaire-based surveys, with a signature or thumbprint. In cases of illiterate participants, DM read the information sheet and written consent form aloud in their local language. These participants were then asked to confirm their understanding, provide verbal consent voluntarily and provide a thumbprint. Written consent forms were obtained from the caregivers of the observed paediatric patients (critically ill children) who were observed. Participants were informed that confidentiality would be maintained, no personal details would be divulged, their involvement was voluntary, and that they could withdraw at any time without consequence.

## Results

### Respondent characteristics

**In-depth interviews.** In total, 23 in-depth interviews (IDIs) were conducted at Zomba Central Hospital's paediatric HDU, with healthcare providers (n = 14) and caregivers (n = 9) of nine critically ill children (aged 3 months to 5 years old)

admitted to the HDU. Of the 14 healthcare providers interviewed, 7 were nurse midwife technicians, and 5 were nursing officers. Most healthcare providers were female (n = 10). The nine caregivers interviewed were all female and mostly aged between 20 and 30 years (n = 6). S1 Table gives a detailed overview of IDI respondent characteristics.

## Questionnaire-based survey

In total, 96 questionnaires were administered in the paediatric HDU at Zomba Central Hospital, involving healthcare providers (n = 24) and caregivers (n = 72) of critically ill children (aged 3 months to 5 years) admitted to the HDU. Of 24 healthcare providers, 13 were nurse midwife technicians, and 6 were nursing officers (S2 Table). Providers were predominantly female (n = 17) and had an average age of 33.4 years. Their medical experience was approximately evenly distributed over 1–5 years, 6–10 years and 11 + years. The majority (n = 17) had diplomas in nursing and midwifery, while seven provider respondents had bachelor's degrees in nursing and midwifery.

S3 Table shows the characteristics of the caregiver respondents and their children. Caregivers' and children's mean ages were 28.7 years and 5.4 months, respectively. The reasons for admission were respiratory (54%), neurological (33.3%), circulatory (6%), sepsis (6%), and gastrointestinal problems (1%). Of the 72 caregivers, 96% were females. Most (65%) caregivers were illiterate or did not finish primary school; 17% and 18% had completed primary and secondary school, respectively. Fig 4 summarises the relationship between the CFIR domains and associated constructs, how they influenced the use of the IMPALA monitor, and its implications for care.

This section will first describe how healthcare providers and caregivers utilise the IMPALA monitor and how it impacts their workload. Next, it will analyse how the following domains and their associated characteristics affect use: intervention characteristics (design quality), outer setting (patient needs and resources), inner setting (access to knowledge and

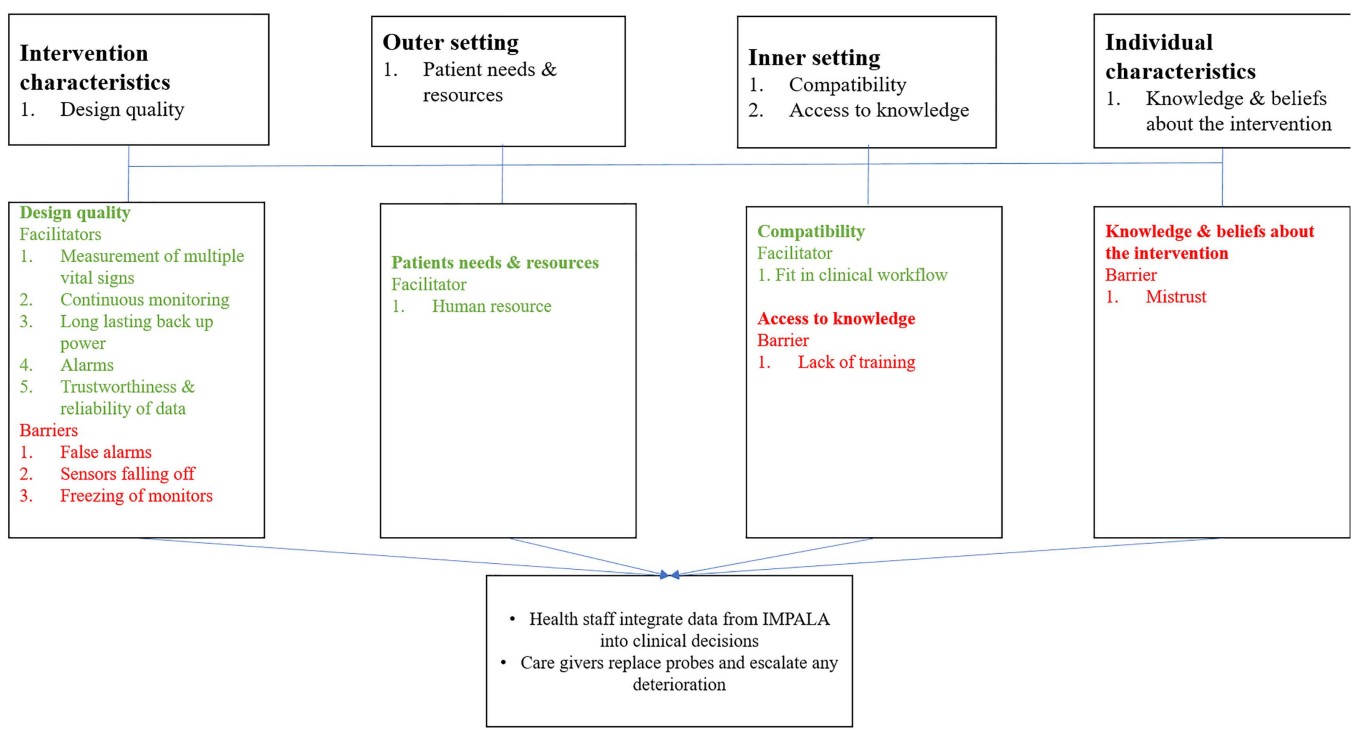

**Fig 4. Relationship of the selected determinants influencing monitor usage and impact on outcome.**

compatibility) and characteristics of individuals (mistrust). Fig 4 further shows the selected domains and their associated constructs or determinants that influenced the use of the monitors.

## Use of the IMPALA monitors and their impact on care

### Using the IMPALA monitor to support clinical decision-making

Healthcare providers reported that using the IMPALA system had a significant impact on clinical decision-making. For instance, it aided in weaning children off oxygen therapy and determining when to discharge stable children based on the vital signs displayed on the monitor. They found that quickly glancing at the monitors allowed them to assess whether a child was deteriorating, facilitating early preventive interventions. By monitoring vital signs, healthcare providers could quickly identify if a child was desaturating and respond by increasing oxygen delivery or checking for loose connections in the monitor probe or oxygen supply. Caregivers also indicated that they could interpret the monitor's colour-coded alarm signals, allowing them to recognise when their child's condition was worsening or if the monitor had become detached. This enabled them to alert the nurse or reattach the probe when necessary.

> *"We are pleased with the intervention because it makes our jobs easier. The IMPALA monitors are attached to the patient, rather than the old monitor, which cannot be used to continuously monitor patients. Therefore, when the nurses enter the HDU, they don't have a lot of work to do in terms of documenting and interpreting vital signs on the monitors to guide them on what action should be undertaken. So, everyone is happy working in the HDU because of this"* (Nurse IDI 08).

Furthermore, healthcare providers and caregivers reported that the alarms alerted them to any deterioration in the patient's condition. This triggered caregivers to flag any changes in the child's condition based on the monitor readings and report them to healthcare providers for possible intervention when a child was unstable, thereby easing the clinical workflow. Most caregivers reported being asked to monitor their child at night and inform healthcare providers (HCPs) of any changes, with 93.1% (95% CI: 87.1–99.1) reporting this either sometimes or always (Table 1). Daytime involvement was slightly lower, with 74.3% (95% CI: 63.8–84.8) reporting similar levels of engagement. Among caregivers, 84.7% (95% CI: 76.2–93.2) reported knowing the correct reporting channel for escalating concerns about their child's condition. Most caregivers perceived HCPs as responsive to changes in their child's condition, with 98.4% (95% CI: 95.1–100) stating that staff "sometimes" or "always" responded quickly, and 88.5% (95% CI: 80.2–96.8) reporting that HCPs "always" responded. Just over half of the respondents (51.3%, 95% CI: 39.6–63.2) reported that an alarm had sounded at some point during their child's HDU stay. Over half, 51.3% of the caregivers (95% CI: 39.6-63-2), always or sometimes heard an alarm sounding during the HDU stay. Among caregivers who had experienced an alarm sounding (n = 38), 94.7% (95% CI: 87.3–100) said HCPs sometimes or always responded promptly, while 73.7% (95% CI: 59.0–88.4) reported that they "always" responded (Table 1).

Caregivers ensured that the monitors continued operating as intended; they were regularly observed reconnecting detached oxygen saturation sensors. However, only some caregivers attempted to reconnect the electrocardiography (ECG) sensors because it was challenging. Any detachment of the ECGs was reported to the nurses so they could intervene. Additionally, caregivers were often observed sleeping, eating, and sitting at the bedside. Their familiarity with their children's conditions before falling ill allowed them to track changes over time, identify acute changes, and escalate concerns to nurses.

> *"My role was to ensure the wires (probes) were connected and not removed. For example, when I was changing my child's sleeping position, I made sure not to remove the wires (probes). If they do, I ensured I had properly put them back, especially for the air (oxygen), so the readings should show on the screen". If I noticed any changes in my child's illness, I would report to the nurses (Caregiver IDI 09)*

**Table 1. The role of caregivers and healthcare providers in monitoring: Caregivers' perspective.**

| Variables | Number of obser-vations (N) | Proportions (%) | 95% CI (%) |
|---|---|---|---|
| Caregiver asked for day monitoring and inform HCPs[a] for any changes (Always or sometimes) (%) | 70 | 74.3 | 63.8, 84.8 |
| Caregiver asked for night monitoring and inform HCPs for any changes (Always or sometimes) (%) | 72 | 93.1 | 87.1, 99.1 |
| Caregivers knowledge on reporting channel for any changes in child condition (Always or sometimes) (%) | 72 | 84.7 | 76.2, 93.2 |
| HCP's respond quickly to changes in child condition[b] (Always or sometimes) (%) | 61 | 98.4 | 95.1, 100 |
| *HCP's respond quickly to changes in child condition[b] (Always) (%)* | 61 | 88.5 | 80.2, 96.8 |
| Alarm sounding during the HDU Stay (Always or sometimes) (%) | 72 | 51.3 | 39.6, 63.2 |
| HCP's respond quickly to alarms[c] (Always or sometimes) (%) | 38 | 94.7 | 87.3, 100 |
| *HCP's respond quickly to alarms[c] (Always) (%)* | 38 | 73.7 | 59.0, 88.4 |

The questionnaire-based caregiver survey (n = 72).

[a] For n = 2, this question was not applicable (not in HDU during the day).

[b] For n = 11, this was not applicable (no changes in condition), and,

[c] for n = 34, this was not applicable (no alarms sounding).

## Impact on the workload of healthcare providers

Healthcare providers reported that the IMPALA monitors reduced their workload. With adequately and continuously functioning monitors, nurses and clinical staff spent less time searching for mobile monitors and pulse oximeters. This alleviated staff fatigue and eased the pressure of conducting one-on-one vital sign checks, which were perceived as time-consuming and too demanding.

> *"They are essential in the HDU regarding the setup because we easily monitor critically ill children. Instead of manually counting vital signs, the IMPALA monitors automatically display the readings. The monitors quickly notify us when the child's condition or vital signs are abnormal for fast intervention. Additionally, the work is simplified because you don't need to use the timer and mobile monitors to check all the children individually".* (Clinician IDI 10)

The healthcare provider questionnaire-based surveys echo this finding. On average, staff reported spending 1.8 hours (95% CI: 1.19–2.48) on child monitoring on a typical day during the IMPALA implementation pilot, compared to 3.3 hours (95% CI: 2.36–4.23) before the implementation of IMPALA, a statistically significant difference (p-value < 0.001). Table 2 lists the tasks that HDU staff identified as their top 3 most time-consuming tasks on a typical day. Admission triage was most frequently cited by 66.7% (95% CI: 46.3–87.0) of the respondents. Completing patient charts was reported by 4.2% (95% CI: 44.5–12.8) of respondents, while 20.8% (95% CI: 3.3–38.4) of respondents mentioned taking vital signs, as well as attending emergencies, cannula insertion, and drug administration to be in their top-3 of most time-consuming tasks. Parental counselling was cited by 16.7% (95% CI: 0.6–32.7), and discharging was cited only 4.2% (95% CI: −4.5–12.8).

## Determinants of monitor use

### Determinants of monitor use at the intervention level

The healthcare providers and caregivers identified several critical factors in the design quality of the IMPALA monitor that contributed to its use. These design quality factors included:

**Table 2.** *Most time-consuming tasks on a typical day for HDU staff.*

| Variables | Number of observations (N) | Proportions (%) | 95% CI (%) |
|---|---|---|---|
| Admission triage | 24 | 66.7 | 46.3, 87.0 |
| Completing patient charts | 24 | 41.7 | 20.4, 62.9 |
| Taking vital signs | 24 | 20.8 | 3.3, 38.4 |
| Attending emergencies | 24 | 20.8 | 3.3, 38.4 |
| Cannula insertion | 24 | 20.8 | 3.3, 38.4 |
| Drug administration | 24 | 20.8 | 3.3, 38.4 |
| Parents counseling | 24 | 16.7 | 0.6, 32.7 |
| Discharge | 24 | 4.2 | − 4.5, 12.8 |

Questionnaire-based healthcare provider survey (n = 24). Respondents were asked to tick at most three tasks in response to the question: "On a typical day, what are your most time-consuming tasks?" with potential answer categories as described above, plus an "Other" category.

**Continuous measurement of multiple vital signs.** Healthcare providers identified key aspects of the IMPALA monitor's design quality that support its use. A major strength was the continuous measurement of multiple vital signs, including heart rate, respiratory rate, oxygen saturation, and blood pressure.. Healthcare providers appreciated that these parameters were readily accessible during clinical assessments and ward rounds. The monitors' wall-mounted setup and the ability to remain continuously connected to critically ill children further enhanced ease of use and supported ongoing monitoring throughout the day and night.

*"We can assess the condition of the patients very easily by looking at the parameters like the heart rate, respiratory rate, oxygen saturation, and blood pressure. Even at a glance at the monitors, you can tell this patient is saturating or desaturating. In addition, it allows you to make decisions in good time".* (Nurse In charge IDI 06)

However, participants expressed concerns that the need for temperature sensors on the IMPALA monitors was a limiting factor in measuring vital signs.

*"I have noticed that the IMPALA monitors do not have temperature sensors, which would be very important. We monitor patients' temperature, and it will be good to have it on the monitor, boosting the IMPALA monitors' usage".* (Nurse IDI 01)

**Easy to read.** The healthcare providers described the IMPALA monitors as very easy to read due to the colours (blue, yellow, and red) and the minimal amount of distracting information. They could quickly and easily collect data at any time without wasting time searching for the monitors or attaching children to them.

*"The IMPALA monitor is easier to use because it is stationary, uncomplicated & has colours such as red, yellow and green and alarms to alert you when the condition of a child is deteriorating. We can use it anytime we want, unlike the other mobile ones. In addition, we can monitor the children well without any interruptions".* (Nurse IDI 07)

**Long-lasting backup power (up to 4 hours).** Healthcare providers appreciated IMPALA's built-in backup battery, which provides up to four hours of power during outages. In the HDU ward, occasional voltage fluctuations did not disrupt monitoring, as the device continued to function seamlessly. In contrast, conventional monitors, which rely solely on external power, ceased operating during power cuts, making IMPALA more reliable in this setting.

*"With the IMPALA monitor, there is continuous monitoring even when there is a power cut, and they keep memories, unlike the mobile monitors where there is intermittent monitoring"* (Nurse IDI 01)

**Alarms.** Healthcare providers and caregivers viewed the alarm feature as crucial in the clinical workflow. Alarms enabled early detection of patient deterioration, prompting timely intervention. This feature also reduced the time and effort required to locate pulse oximeters or other mobile devices during emergencies, easing the workload of healthcare providers.

*"When the alarm sounded, the measurement for air (oxygen) showed that the oxygen saturation was dropping to 89. So when it happened like this, I called for the nurse, and they could increase the air (oxygen levels). I was calling the nurse to check my child".* (Caregiver IDI 02)

However, false alarms prompted nurses to ignore the monitors when they beeped, as they had challenges distinguishing between false alarms and real alarms. Indeed, approximately half (49%) of the caregivers in the survey reported that they did not hear the alarms sounding during their child's stay on the HDU (Table 1). During observations, it was also noted that the nurses changed the alarm thresholds to high or low numbers or turned off the alarm to minimise beeping.

*"I feel like the false alarms should not be there because they make us busy by making us check the monitors occasionally. We could spend the whole day in the HDU trying to check the monitors, which was tiresome on our job when they made the sound, especially when more patients were connected. Therefore, the alarms should beep only when the patient is deteriorating or the probes have disconnected (Study nurse IDI 14).*

**Reliability of measurements.** The IMPALA monitors increased confidence among healthcare providers by providing accurate, reliable vital sign data to support clinical decision-making, such as adjusting oxygen delivery and determining the child's condition. Nevertheless, the healthcare providers emphasised the continued need to apply clinical expertise and to interpret and monitor outputs in conjunction with the patient's present clinical presentation.

*"I think it can work anywhere because it is a monitoring machine, and it's effective. It shows all parameters like BP pulse rate, respiration, and oxygen saturation; these are the parameters we look for in a patient. These parameters serve as baseline data to help us determine the appropriate action to take. So, I believe it can work in all the HDUs."* (Clinician IDI 09).

However, technical issues such as sensors falling off, the absence of temperature sensors, false readings due to child movement, monitor freezing, and incorrect high blood pressure values undermined the monitors' reliability. This led to decreased engagement with the device among clinical staff.

*"The disadvantage of the IMPALA monitors is that the sensors or probes easily get disconnected when the child is too mobile, leading to false results. This affects our work, and sometimes we lose trust in the monitors. Lack of temperature sensor is also a challenge that IMPALA monitors have, affecting our work".* (Nurse IDI 07)

## Determinants of monitor use at the Outer setting level

Healthcare providers identified patient needs and human resources as critical factors contributing to IMPALA's usage.

**Human resource.** Healthcare providers reported that the availability of IMPALA monitors simplified the task of assessing and collecting vital signs for multiple critically ill children (eight to twelve critically ill children), often managed by a single nurse. The monitors enabled quick identification and prioritisation of patients in need of urgent care, particularly during assessments and ward rounds. The healthcare provider questionnaire-based surveys echo this finding. Staff reported spending less time on monitoring children, 1.8 hours daily during the IMPALA implementation pilot, compared to 3.3 hours pre-IMPALA.

*"Firstly, it eases our work, especially nurses, since we are few. So, assessing the patients individually becomes a challenge. It happens that when you are busy attending to one patient, you might not be aware of how critical the other patient is and in need of haste treatment; hence, with the IMPALA monitor, it helps to prioritise which patient is to be attended to first depending on how critical the patient is".* (Nurse IDI 08)

## Determinants of monitor use at the Inner setting level

Healthcare providers identified two factors at this level that influence the uptake and use of the IMPALA, which monitors compatibility and readiness for implementation, as well as access to knowledge.

**Fit in clinical workflow (compatibility).** Healthcare providers and caregivers reported that IMPALA monitors integrated well into clinical workflows by providing real-time, reliable, and non-invasive vital signs data. This continuous monitoring supported timely decision-making, such as discharging stable patients. It enabled more efficient management of critically ill children, as nurses could quickly identify changes in the child's condition.

*"What I like about the IMPALA monitors the most is that they ease nurses' work and are not painful when connected to our children. Instead of the nurse moving the monitors to get the vital signs, they record the data on the monitors. There was a child who was not connected to the monitor, so every time the nurse wanted to check the vital signs, they had to move the monitor from another patient to this one to record the data. But with my child, it was easy to collect the data because my child was connected to the monitor throughout".* (Caregiver IDI 07)

**Training on technology.** Healthcare providers reported receiving training on using the IMPALA monitors, including how to interpret vital signs data and connect sensors to patients. This training enhanced their ability to efficiently collect and document vital signs on critical care pathway charts. However, some healthcare providers indicated that the training was limited, particularly in hands-on practice and technical features like navigating the scroll menu. Most relied on informal, on-the-job learning, which was considered insufficient, especially in the context of frequent staff rotations in the HDU. Nurses who joined after the system's introduction often had only brief, once-off orientation sessions and lacked confidence in operating the monitors (connecting or disconnecting children to the monitors). The surveys revealed that 66.7% of healthcare providers had no critical or intensive care training, 95.8% lacked paediatric critical or intensive care training, and 70.8% were not trained in paediatric HDU monitoring before the IMPALA.

*"OK, I think if there are opportunities for briefing or refresher training on IMPALA, they need to include people who were left out during the previous briefing and staff who have recently just joined. If that happens, it will be better because most staff will be knowledgeable"* (Clinician IDI 10)

This lack of training hindered some healthcare providers from effectively setting up and adjusting the settings on the IMPALA monitor, leading to a lack of confidence in its use and the inability to connect the children to the monitors.

## Determinants of monitor use related to the knowledge and beliefs of individuals

Healthcare providers and caregivers identified knowledge and beliefs about the intervention (i.e., individual attitudes toward and the value placed on the intervention, as well as familiarity with facts, truths, and principles related to the intervention) as a factor influencing the uptake and use of the IMPALA monitors.

**Trust and mistrust.** Some caregivers perceived the monitors as helpful in detecting a child's prognosis and beneficial for escalating deterioration to healthcare providers. They considered the availability of different colours (green, yellow and red) helpful in alerting and informing about the child's progress. Four out of nine female of caregivers interviewed believed

that connecting critically ill children to these devices could worsen their condition and lead to death. They associated the continuous monitoring with superstitions and fears, holding the misconception that children connected to these devices were more likely to die. They reported that the most severely ill children on the monitors had a higher likelihood of dying. This belief extended to the misconception that oxygen therapy, when combined with the monitors, was a way of hastening a child's death, which intensified their anxiety and reluctance to participate in the study.

*"Other people said that IMPALA is destroying children. They claim that this causes children not to recover quickly or, even when they do, they do not wait long before returning to the HDU for admission and die".* (Caregiver IDI 09).

However, when such caregivers were asked about these beliefs, most acknowledged their fears about the potential adverse effects of using the monitors. Still, they found it difficult to align their concerns with the evidence that indicates that monitors may harm children. Over time, as they observed the monitors' positive impact on the quality of patient care during admissions, their perceptions began to shift. This helped to dispel their initial fears and facilitated their acceptance of the devices.

*"At first, I thought that the monitors were dangerous and could kill my child, but I noted that the monitors did not bring any harm to the child and helped my child to receive good care and support during hospitalisation".* (Caregiver IDI 02)

## Discussion

This study is the first to employ a mixed-methods implementation research approach to scientifically examine barriers and facilitators of effectively implementing a continuous monitoring system in paediatric critical care in an LMIC setting. All healthcare providers and caregivers recognised the value of continuous vital signs monitoring for enabling timely responses to clinical deterioration in children.

Key design features of the IMPALA monitor include the simultaneous measurement of multiple vital signs, clear readability, long-lasting backup power, and an integrated alarm system, all of which enhance its compatibility with existing systems. These features streamlined care by consolidating previously fragmented tasks and enabled real-time monitoring, which facilitated timely clinical interventions, reduced nurses' workload, and improved clinical decision-making and health outcomes [19–21]. Before implementation, nurses often juggled multiple devices (mobile monitors and pulse oximeters) to gather vital signs such as heart rate, blood pressure, respiratory rate, and oxygen saturation. This intermittent monitoring delayed the recognition of deterioration, particularly during busy shifts, and hindered timely decisions [22,23]. IMPALA simplified workflows by integrating these functions into a single device, thereby saving time and enabling more direct patient care. However, its effectiveness was constrained by staff shortages and the high workload in the paediatric HDU. Fig 2 underscores how clinical decisions were often shaped as much by the environment as by medical protocols.

Despite its usability benefits, the IMPALA system generated a high volume of data, which, when combined with patient records and laboratory results, posed a risk of information overload [24]. Without adequate training, healthcare providers may struggle to interpret and effectively act upon the data. Simplified, easily interpretable summaries are essential, particularly in high-demand settings like the HDU [25–28].

The alarm system was a critical component of usability. It allowed nurses to triage care according to the severity of illness without manually checking every child, thereby reducing unnecessary diagnostic tests and promoting more efficient resource allocation in resource-limited settings [29]. Alarms facilitated early detection of deterioration, improving care quality and clinical outcomes. [30,31].

However, frequent false alarms disrupted workflows and contributed to alarm fatigue, as observed in other intensive care settings [32]. Alarm fatigue can desensitise healthcare providers, increase stress, burnout, and compromise patient safety [32–35]. Addressing this requires customisable alarm settings, integration of alarms into clinical workflows, ongoing

training, routine feedback audits, and the development of clear alarm management protocols [36–43]. The IMPALA team is currently developing supporting algorithms to refine, monitor outputs, and tailor alarm settings.

Training emerged as a significant barrier to effective use. Nurses reported limited confidence in setting up the monitors, connecting critically ill children, interpreting alarms, and reviewing patient histories. Hospital staff had less hands-on experience and received minimal formal training compared to study nurses. A previous qualitative study at the same hospital identified the need for ongoing staff training on available monitors, particularly for new staff working in HDUs [1,44]. One-off training was insufficient; continuous training, ideally delivered by local supervisors or through e-learning platforms, must be embedded into implementation strategies. Monitoring itself does not improve outcomes unless providers can respond appropriately to the information.. Training should reinforce basic paediatric nursing skills, recognition of age-specific vital signs and interpretation of physiological parameters to improve paediatric critical care in HDUs [2]. Improved training can also strengthen communication between nurses and caregivers, reduce caregiver mistrust, foster their active involvement in monitoring, and alleviate the workload of clinical staff [45].

Trust and mistrust in the IMPALA monitors also shaped the effective usage. Many caregivers viewed the device as a helpful tool for assessing their child's condition and alerting healthcare providers to deterioration. Studies have shown that family-initiated escalation of care can expedite clinical responses and improve health outcomes [46–48]. However, some caregivers expressed concerns based on cultural beliefs and superstitions.. According to our qualitative data, four out of nine of our female caregivers reported feeling suspicious about the device that connecting critically ill children to the monitor could worsen their condition or lead to death. They observed that children on the monitor were often those who died, which reinforced that the device, in conjunction with oxygen therapy, hastened death [49]. These beliefs heightened caregiver anxiety and contributed to fears regarding the connection of their children to the monitors.

Addressing such beliefs is essential, particularly given caregivers' potential role in overstretched health systems. In LMICs, family members often support care by feeding, cleaning, administering basic treatments, and alerting staff to emergencies [47,50]. However, at Zomba Central Hospital, caregivers' roles remain informal and untrained [1]. Clarifying and supporting these roles could strengthen the contribution of caregivers in critical care delivery, particularly in settings facing chronic staff shortages.

## Strengths and limitations of the study

This study employed a robust mixed-methods design to evaluate the implementation of the IMPALA continuous monitoring system, integrating qualitative and quantitative data to enhance the depth and validity of findings. Triangulation of multiple data sources facilitated a comprehensive understanding of implementation processes. At the same time, sustained engagement in the HDU such as spending hundreds of hours of observation enabled the research team to build rapport and trust with healthcare providers.

Several strategies were implemented to reduce potential biases in qualitative data collection and analysis. While blinding is inherently limited in qualitative research, interviewer bias was mitigated through the use of a standardised, piloted interview guide informed by the Consolidated Framework for Implementation Research (CFIR). This ensured consistency across interviews and minimised suggestive questioning. Two independent analysts conducted iterative coding, engaged in regular peer debriefings, and reached consensus on the coding framework to strengthen analytical rigour and reduce interpretive bias. Reflexivity was actively practised throughout the analysis, with researchers reflecting on their positionality and potential assumptions. All interviews were conducted in participants' preferred languages and professionally translated to minimise interpreter bias.

Nonetheless, the study had several limitations. The relatively small sample size in both qualitative and quantitative components may limit generalisability. We noted that interviewing caregivers after discharge may have introduced selection bias, as those with either more positive or more negative experiences could lead to an increased willingness to participate. However, all the invited caregivers consented to participate, such that selection bias remained limited.

Observer effects may also have influenced participants' behaviour during the HDU observations. While researchers made efforts to minimise disruption and blend into the clinical environment, their presence may still have affected how staff interacted with the monitors. Additionally, caregiver responses may have been influenced by courtesy bias, particularly if gratitude towards healthcare providers shaped their feedback. To mitigate this, participants were assured at the start of each interview that their responses would remain anonymous and would not affect their future access to care.

## Conclusion

The IMPALA monitoring system has demonstrated significant value in improving the management of critically ill children at Zomba Central Hospital. Its ability to simultaneously and continuously monitor multiple vital signs: heart rate, respiratory rate, blood pressure, and oxygen saturation was viewed favourably by both healthcare providers and caregivers. Compared to intermittent monitoring methods, continuous monitoring was perceived as more reliable, less labour-intensive, and better suited to support timely clinical decision-making.

For long-term adoption and potential scale-up of IMPALA in other low-resource settings, several areas for improvement were identified. These include enhancing battery capacity to ensure functionality during frequent power outages, optimising the user interface through ongoing caregiver and provider input, and reducing alarm fatigue through improved alarm management strategies. Ongoing, structured training for healthcare providers is also essential to ensure consistent and competent use of the device.

Successful integration into routine clinical workflows will depend not only on technological improvements but also on broader health system strengthening. Embedding IMPALA within existing critical care protocols, supported by targeted training in paediatric monitoring and clinical response, will be critical for sustained use. Equally important is addressing caregiver trust and misconceptions, particularly fears linking monitoring devices to child mortality. Promoting transparent communication, clarifying caregivers' roles, and actively engaging them as partners in care can foster trust, reduce anxiety, and alleviate the burden on overstretched clinical staff.

More broadly, as continuous monitoring technologies become increasingly viable for LMICs, these findings offer practical guidance for policymakers, programme implementers, and health system planners. Lessons from this implementation study can inform procurement strategies, integration plans, and community engagement approaches across similar settings. Tailoring implementation to local health system capacity, workforce constraints, and cultural context will be essential to ensuring sustainable adoption and ultimately improving paediatric critical care outcomes across low-resource environments.

## Supporting information

**S1 Table. In-depth interviews respondent characteristics.**
(XLSX)

**S2 Table. Healthcare Providers survey participants.**
(XLSX)

**S3 Table. Children and Caregiver characteristics.**
(XLSX)

**S1 File. The IMPALA Study Team†**
(DOCX)

## Acknowledgments

We want to thank our participants and the rest of the ward staff for their valuable time, as we would not have been able to share our findings without them.

## Author contributions

**Conceptualization:** Daniel Mwale, Christopher Pell, Jobiba Chinkhumba, Happiness Mandala, Jessica Chikwana, Josephine Langton, Michael Boele van Hensbroek, IMPALA Study Team, Job Calis, Wendy Janssens, Lucinda Manda-Taylor.

**Data curation:** Daniel Mwale, Happiness Mandala, Alice Likumbo, IMPALA Study Team.

**Formal analysis:** Daniel Mwale, Christopher Pell, Josephine Langton, Michael Boele van Hensbroek, Job Calis, Wendy Janssens, Lucinda Manda-Taylor.

**Funding acquisition:** Job Calis.

**Investigation:** Daniel Mwale, Christopher Pell, Jobiba Chinkhumba, Jessica Chikwana, Josephine Langton, Alice Likumbo, Michael Boele van Hensbroek, IMPALA Study Team, Wendy Janssens, Lucinda Manda-Taylor.

**Methodology:** Daniel Mwale, Christopher Pell, Jobiba Chinkhumba, Jessica Chikwana, Josephine Langton, Michael Boele van Hensbroek, IMPALA Study Team, Job Calis, Wendy Janssens, Lucinda Manda-Taylor.

**Project administration:** Daniel Mwale, Christopher Pell, Jessica Chikwana, Josephine Langton, Alice Likumbo, Michael Boele van Hensbroek, IMPALA Study Team, Job Calis, Wendy Janssens, Lucinda Manda-Taylor.

**Resources:** IMPALA Study Team, Job Calis.

**Software:** Daniel Mwale, Christopher Pell, Jobiba Chinkhumba, Job Calis.

**Supervision:** Daniel Mwale, Christopher Pell, Jobiba Chinkhumba, Josephine Langton, Michael Boele van Hensbroek, IMPALA Study Team, Job Calis, Wendy Janssens.

**Validation:** Daniel Mwale, Happiness Mandala, Jessica Chikwana, Josephine Langton, Alice Likumbo, Michael Boele van Hensbroek, IMPALA Study Team, Job Calis, Lucinda Manda-Taylor.

**Visualization:** Daniel Mwale.

**Writing – original draft:** Daniel Mwale.

**Writing – review & editing:** Daniel Mwale, Christopher Pell, Jobiba Chinkhumba, Happiness Mandala, Jessica Chikwana, Josephine Langton, Alice Likumbo, Michael Boele van Hensbroek, Job Calis, Wendy Janssens, Lucinda Manda-Taylor.

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
