## [Decision Letter · Decision Letter 0]

27 May 2025

Dear Dr. Mwale,

Thank you for submitting your manuscript to PLOS ONE. After careful consideration, we feel that it has merit but does not fully meet PLOS ONE’s publication criteria as it currently stands. Therefore, we invite you to submit a revised version of the manuscript that addresses the points raised during the review process.

We look forward to receiving your revised manuscript.

Kind regards,

Nik Hisamuddin Nik Ab. Rahman

Academic Editor

PLOS ONE

**Journal Requirements:**

1. When submitting your revision, we need you to address these additional requirements. Please ensure that your manuscript meets PLOS ONE's style requirements, including those for file naming. The PLOS ONE style templates can be found at https://journals.plos.org/plosone/s/file?id=wjVg/PLOSOne_formatting_sample_main_body.pdf and https://journals.plos.org/plosone/s/file?id=ba62/PLOSOne_formatting_sample_title_authors_affiliations.pdf 2. Thank you for stating in your Funding Statement: This project is part of the European and Developing Countries Clinical Trials Partnership 2 programme supported by the European Union (Grant number RIA20201-3294 – IMPALA). The grant was awarded to JC. The funders had no role in study design, data collection and analysis, decision to publish, or preparation of the manuscript. edctp.org  Please provide an amended statement that declares *all* the funding or sources of support (whether external or internal to your organization) received during this study, as detailed online in our guide for authors at http://journals.plos.org/plosone/s/submit-now.  Please also include the statement “There was no additional external funding received for this study.” in your updated Funding Statement. Please include your amended Funding Statement within your cover letter. We will change the online submission form on your behalf. 3. Please know it is PLOS ONE policy for corresponding authors to declare, on behalf of all authors, all potential competing interests for the purposes of transparency. PLOS defines a competing interest as anything that interferes with, or could reasonably be perceived as interfering with, the full and objective presentation, peer review, editorial decision-making, or publication of research or non-research articles submitted to one of the journals. Competing interests can be financial or non-financial, professional, or personal. Competing interests can arise in relationship to an organization or another person. Please follow this link to our website for more details on competing interests: http://journals.plos.org/plosone/s/competing-interests 4. We note that you have indicated that there are restrictions to data sharing for this study. For studies involving human research participant data or other sensitive data, we encourage authors to share de-identified or anonymized data. However, when data cannot be publicly shared for ethical reasons, we allow authors to make their data sets available upon request. For information on unacceptable data access restrictions, please see http://journals.plos.org/plosone/s/data-availability#loc-unacceptable-data-access-restrictions.  Before we proceed with your manuscript, please address the following prompts: a) If there are ethical or legal restrictions on sharing a de-identified data set, please explain them in detail (e.g., data contain potentially identifying or sensitive patient information, data are owned by a third-party organization, etc.) and who has imposed them (e.g., a Research Ethics Committee or Institutional Review Board, etc.). Please also provide contact information for a data access committee, ethics committee, or other institutional body to which data requests may be sent. b) If there are no restrictions, please upload the minimal anonymized data set necessary to replicate your study findings to a stable, public repository and provide us with the relevant URLs, DOIs, or accession numbers. Please see http://www.bmj.com/content/340/bmj.c181.long for guidelines on how to de-identify and prepare clinical data for publication. For a list of recommended repositories, please see https://journals.plos.org/plosone/s/recommended-repositories. You also have the option of uploading the data as Supporting Information files, but we would recommend depositing data directly to a data repository if possible. Please update your Data Availability statement in the submission form accordingly. 5. One of the noted authors is a group or consortium “IMPALA consortium”. In addition to naming the author group, please list the individual authors and affiliations within this group in the acknowledgments section of your manuscript. Please also indicate clearly a lead author for this group along with a contact email address. 6. Please include captions for your Supporting Information files at the end of your manuscript, and update any in-text citations to match accordingly. Please see our Supporting Information guidelines for more information: http://journals.plos.org/plosone/s/supporting-information. 7. Please review your reference list to ensure that it is complete and correct. If you have cited papers that have been retracted, please include the rationale for doing so in the manuscript text, or remove these references and replace them with relevant current references. Any changes to the reference list should be mentioned in the rebuttal letter that accompanies your revised manuscript. If you need to cite a retracted article, indicate the article’s retracted status in the References list and also include a citation and full reference for the retraction notice.

Reviewers' comments:

**Comments to the Author**

1. Is the manuscript technically sound, and do the data support the conclusions?

Reviewer #1: Yes

Reviewer #2: Yes

Reviewer #3: Yes

Reviewer #4: Yes

2. Has the statistical analysis been performed appropriately and rigorously?

Reviewer #1: Yes

Reviewer #2: Yes

Reviewer #3: Yes

Reviewer #4: Yes

3. Have the authors made all data underlying the findings in their manuscript fully available?

Reviewer #1: Yes

Reviewer #2: Yes

Reviewer #3: Yes

Reviewer #4: Yes

4. Is the manuscript presented in an intelligible fashion and written in standard English?

Reviewer #1: No

Reviewer #2: Yes

Reviewer #3: Yes

Reviewer #4: No

**Reviewer #1:**  Sampling and Potential Bias:

Please clarify the sampling strategy for caregivers and healthcare providers to ensure transparency and highlight the representativeness of the study population.

Consider discussing potential selection and courtesy bias in caregiver responses, especially since interviews were conducted post-discharge.

Clarity and Language:

A thorough language review is recommended to correct grammatical inconsistencies, repetitive phrasing, and tense shifts.

Please revise long or redundant sentences, particularly in the Discussion section, to improve readability.

Ethical Procedures:

Kindly specify who conducted the informed consent process and how it was administered, especially for illiterate participants.

Data Availability Statement:

The current statement indicates that data are available upon request due to confidentiality. In line with PLOS ONE's open data policy, I recommend that the authors de-identify and publicly share applicable quantitative datasets or provide a clear justification for the restriction.

Statistical Analysis:

Where appropriate, please consider including confidence intervals or p-values to give more context to the descriptive statistics.

Clarify whether any comparative analyses were performed (e.g., before vs. after implementation, or between trained vs. untrained staff), even if exploratory.

Policy Implications and Scalability:

Consider expanding the conclusion to highlight how these findings can inform broader implementation or scale-up in other low-resource settings.

Monitor Design and Alarm Features:

Given the recurring issue of false alarms and alarm fatigue, please elaborate on whether alarm thresholds or settings were adjustable and how this will be addressed in future iterations of the IMPALA monitor.

Visual Aids and Figures:

Please ensure that all figures (e.g., floor plan, monitor diagram) are of publication quality, with clear labeling and resolution.

Non-response in Survey Data:

In Table 1, nearly half of the caregivers did not respond to some items related to alarm response. Please address this non-response and discuss its potential implications for interpretation of the data.

**Reviewer #2:**  This is an important and well-structured study that addresses a significant gap in pediatric critical care in low-resource settings. The mixed-methods approach is appropriate and strengthens the depth and credibility of the findings. The study clearly outlines both barriers and facilitators to implementing the IMPALA monitoring system in a real-world hospital environment in Malawi.

The qualitative component is well-executed, with thoughtful use of the CFIR framework and rich insights from both caregivers and healthcare providers. The quantitative data are clearly presented and appropriately analyzed. Together, the findings support the authors’ conclusions about the potential benefits of the IMPALA system, especially in improving clinical decision-making, reducing staff workload, and involving caregivers more effectively in monitoring.

There are just a few minor language and typographical errors that can be addressed at the copyediting stage (e.g., a few spelling inconsistencies like “verson” instead of “version”). Otherwise, the manuscript is written in clear and standard English.

Overall, this is a well-conducted implementation study with important implications for pediatric care in similar low-resource settings, and I support its publication.

**Reviewer #3:**  Dear Author,

Thank you for your submission.

The authors assessed the implementation barriers and facilitators of a locally adapted,

robust, low-cost continuous monitoring system (IMPALA) in Malawi. The study used a mixed-methods design. Including healthcare provider and caregiver perspectives enriches the contextual understanding of implementation barriers and facilitators. Nevertheless, there are some aspects that can be improved for the readability of the paper. Some typo errors should be corrected.

Additional comments:

1- Introduction:

• The researchers are encouraged to add some statistics from the literature regarding morbidity/ mortality of children in critical care units in Malway or other relevant healthcare contexts. This could further clarify the research problem

2- Method and setting

Design:

• The researchers are recommended to report the reason for selecting the mixed method design and explain how it was useful in investigating the research problem.

Data collection:

• The researchers are encouraged to attach the two structured observation guides

The questionnaire:

- The researchers are recommended to report information about the validity and reliability of the self-administered EMPATHIC-65 questionnaire.

Intervention:

• The researchers should add more information to support the novelty of the IMPALA Continuous Monitoring System.

Quantitative Data:

• The researchers did not use inferential statistics. Such statistical tests could help to explain the credibility of the intervention.

Sample size:

• The sample size is recommended to be calculated for the statistical test used to ensure the power, ES, and alpha level of significance are appropriate for the reported sample size.

4. Results:

• The quality of the figures should be improved regarding clarity and resolution.

Sincerely,

**Reviewer #4:**  The manuscript presents an important and timely study evaluating the implementation of the IMPALA monitoring system for paediatric critical care in a low-resource setting. The use of a mixed-methods design, guided by the CFIR framework, is appropriate and methodologically sound. The authors have made an impressive effort in collecting and analysing a large amount of qualitative and quantitative data. The integration of multiple perspectives, including caregivers and frontline healthcare providers, is a strength of the study.

However, several revisions are recommended to enhance the clarity, conciseness, and overall impact of the manuscript:

Abstract:

The abstract is overloaded with numeric details (e.g., 300+ hours of observation, n=14, n=72) which detract from the clarity of the findings. The text would benefit from a more concise presentation using compact summaries, such as:

“Using over 300 hours of observations, interviews with 14 providers and 9 caregivers, and surveys from 24 providers and 72 caregivers…”

Introduction:

The introduction could benefit from a more technically specific description of the IMPALA system. For example, outlining the technical advantages, usability features, or target age group would help better position the device within the paediatric care landscape.

Methods:

The methods section is comprehensive and demonstrates strong methodological planning. The use of multiple data sources and triangulation is a clear strength.

However, the section is too lengthy and contains some redundancies. For example, the detailed training procedures on IMPALA use could be summarized and partially moved to the Results or Supplementary Materials.

References to figures (especially Figure 2) are underdeveloped. More context should be provided on how the physical setup of the HDU (e.g., multiple children sharing beds, limited equipment) influenced monitoring practices.

While the software used for analysis is clearly stated (NVivo, STATA), the manuscript should briefly describe how potential biases in qualitative interpretation (e.g., lack of blinding, interpreter bias) were mitigated.

Results:

The results section is informative but overly long and repetitive. Several points (e.g., alarm features, technical issues) are discussed multiple times. The use of clear subheadings based on the CFIR domains is commendable but should be streamlined for brevity.

A significant theme—caregiver mistrust and superstition (e.g., “IMPALA kills children”)—is only superficially covered. Its prevalence, distribution by caregiver demographics, and potential impact on implementation should be further explored or quantified.

**Do you want your identity to be public for this peer review?** For information about this choice, including consent withdrawal, please see our Privacy Policy

Reviewer #1: No

Reviewer #2: **Yes: ** Abdullah Abdullah Abbas Al-Murad

Reviewer #3: **Yes: ** Saleh Al Omar

Reviewer #4: No

---

## [Author Response · Author response to Decision Letter 1]

10 Jul 2025

Mr Daniel Mwale

Kamuzu University of Health Sciences

P/Bag 360

Chichiri

Blantyre 3

Email address: dmwale@kuhes.ac.mw

08th July 2025

Reviewers

PLOS ONE

Dear Editor(s),

We thank you for the time you spent assessing our manuscript, for your careful reading, and for your suggestions and recommendations. Your current feedback has helped us to improve the quality of this paper. In revising our manuscript with reference submission ID PONE-D25-13594 entitled “Implementing the IMPALA continuous monitoring system for paediatric critical care in Malawi: A mixed methods study of barriers and facilitators”, we have carefully considered all the comments and suggestions and made necessary revisions.

This research article describes the implementation barriers and facilitators of a locally adapted, robust, low-cost continuous monitoring system (IMPALA) in Malawi. This is one of the first studies that specifically examines monitoring practices for paediatrics in sub-Saharan Africa. IMPALA monitors enhance care by continuously and reliably measuring multiple vital signs, enabling healthcare staff to respond immediately to any deterioration in vital signs. The monitors allow healthcare providers to spend less time monitoring children (1.8 hours daily compared to 3.3 hours before IMPALA). However, alarm fatigue, limitations in knowledge of the technology, and staff shortages were recognised as barriers to IMPALA’s use.

Following your comments, we have incorporated all the requested information into the manuscript, as highlighted in the track changes. We have also provided a point-by-point response to the Reviewer as well as the editors.

As requested, the financial disclosure statement amendment has been revised as follows: “The European and Developing Countries Clinical Trials Partnership 2 programme supported by the European Union (Grant number RIA20201-3294 – IMPALA) provided all the funding or sources of support. There was no additional external funding received for this study. The grant was awarded to JC. The funders had no role in study design, data collection and analysis, decision to publish, or preparation of the manuscript. edctp.org”.

We thank you for considering our manuscript and look forward to your response.

Kind regards

Daniel Mwale

Detailed response to editor comments:

1. Editor's comment: When submitting your revision, we need you to address these additional requirements.

Response: Thank you for your comment. The manuscript has been revised to meet the additional requirements

2. Editor's comment: Thank you for stating in your Funding Statement:

This project is part of the European and Developing Countries Clinical Trials Partnership 2 programme supported by the European Union (Grant number RIA20201-3294 – IMPALA). The grant was awarded to JC. The funders had no role in study design, data collection and analysis, decision to publish, or preparation of the manuscript. edctp.org. Please provide an amended statement that declares *all* the funding or sources of support (whether external or internal to your organization) received during this study, as detailed online in our guide for authors at http://journals.plos.org/plosone/s/submit-now. Please also include the statement “There was no additional external funding received for this study.” in your updated Funding Statement.

Response: Thank you for raising important point. We have revised the funding statement as follows:

The European and Developing Countries Clinical Trials Partnership 2 programme supported by the European Union (Grant number RIA20201-3294 – IMPALA) provided all the funding or sources of support. There was no additional external funding received for this study. The grant was awarded to JC. The funders had no role in study design, data collection and analysis, decision to publish, or preparation of the manuscript. edctp.org.

3. Editors comment: Please know it is PLOS ONE policy for corresponding authors to declare, on behalf of all authors, all potential competing interests for the purposes of transparency. PLOS defines a competing interest as anything that interferes with, or could reasonably be perceived as interfering with, the full and objective presentation, peer review, editorial decision-making, or publication of research or non-research articles submitted to one of the journals. Competing interests can be financial or non-financial, professional, or personal. Competing interests can arise in relationship to an organization or another person. Please follow this link to our website for more details on competing interests: http://journals.plos.org/plosone/s/competing-interests.

Response: Thank you for your comment. We noted that the competing interest statement was submitted during the initial submission. However, the competing interest declaration is as follows:

The authors declare that they have no known competing financial interests or personal relationships that could have appeared to influence the work reported in this paper. The authors declare the following financial interests/personal relationships, which may be considered as potential competing interests; financial support, administrative support, equipment, supplies, travel, and writing assistance were provided by Kamuzu University of Health Sciences.

Editor's comment: We note that you have indicated that there are restrictions to data sharing for this study. For studies involving human research participant data or other sensitive data, we encourage authors to share de-identified or anonymized data. However, when data cannot be publicly shared for ethical reasons, we allow authors to make their data sets available upon request. For information on unacceptable data access restrictions, please see http://journals.plos.org/plosone/s/data-availability#loc-unacceptable-data-access-restrictions.

Response: Thank you for rasining this important comment. We have have indicated that there are no restrictions to data sharing in the portal.

Response: Thank you for the comment. We acknowledge that the de-identified data set will be shared without any restrictions.

b) If there are no restrictions, please upload the minimal anonymized data set necessary to replicate your study findings to a stable, public repository and provide us with the relevant URLs, DOIs, or accession numbers. Please see http://www.bmj.com/content/340/bmj.c181.long for guidelines on how to de-identify and prepare clinical data for publication. For a list of recommended repositories, please see https://journals.plos.org/plosone/s/recommended-repositories. You also have the option of uploading the data as Supporting Information files, but we would recommend depositing data directly to a data repository if possible. Please update your Data Availability statement in the submission form accordingly.

Response: Thank you for raising this important point. Anonymised data set will be shared to replicate the study findings and will be shared as supporting information. Data availability statement in the submission form has been revised accordingly.

5. Edtor's comment: One of the noted authors is a group or consortium “IMPALA consortium”. In addition to naming the author group, please list the individual authors and affiliations within this group in the acknowledgments section of your manuscript. Please also indicate clearly a lead author for this group along with a contact email address.

Response: Thank you for your valuable comment. We have revised the "IMPALA consortium" as an author in the acknowledgement section of the manuscript. We have clearly indicated a lead author for the group along with a contact email address.

6. Editor's comment:Please include captions for your Supporting Information files at the end of your manuscript, and update any in-text citations to match accordingly. Please see our Supporting Information guidelines for more information: http://journals.plos.org/plosone/s/supporting-information.

Response: Thank you for your interesting comment. We have included the captions for the supporting information files at the end of the manuscript and the in-text citations to match accordingly.

7. Editor's comment: Please review your reference list to ensure that it is complete and correct. If you have cited papers that have been retracted, please include the rationale for doing so in the manuscript text, or remove these references and replace them with relevant current references. Any changes to the reference list should be mentioned in the rebuttal letter that accompanies your revised manuscript. If you need to cite a retracted article, indicate the article’s retracted status in the References list and also include a citation and full reference for the retraction notice.

Response: Thank you for your comment. We have checked all the references, the references are correct and complete. The reference list has changed based on the reviewer's comments. The reviewers recommended adding more information from the literature in the introduction and that has been included accordingly.

Detailed responses to Reviewer 1’s comments

Reviewer 1 comments: Please clarify the sampling strategy for caregivers and healthcare providers to ensure transparency and highlight the representativeness of the study population.

Response: Thank you for the suggestion, we have clarified the sampling strategy and included it in the methods section “Respondents sampling and recruitment”. Specifically for the qualitative data, the plan was as follows:

In total, 23 observations and in-depth interviews (IDIs) were conducted at Zomba Central Hospital's paediatric HDU, with healthcare providers (n = 14) and caregivers (n = 9) of nine critically ill children (aged 3 months to 5 years old) admitted to the HDU. After introducing the IMPALA monitors, study participants (healthcare providers and caregivers) were conveniently recruited for structured observations and in-depth interviews between November 9, 2022, and October 30, 2023. Healthcare providers of different cadres (nurses, clinical officers and medical doctors) working in HDU were recruited (face-to-face) during their day shifts, representing a mix of professional roles and levels of experience. Selection was based on their involvement in paediatric care, treatment and management. To enhance diversity and representativeness, caregivers of critically ill children with a range of diagnoses and socioeconomic status were also recruited during their admission to HDU. (Refer to lines 223-234)

Caregivers were selected based on being above eighteen years of age and fulfilling a caregiving role. This approach ensured capturing the lived experience of providing care for critically ill children (refer to lines 240-242)

For the quantitative data, the strategy was as follows:

The sampling frame for the questionnaire-based caregiver survey consisted of all caregivers of critically ill children aged three months to five years who were admitted to the HDU between January 1 and June 30, 2023, and consented to participate in the IMPALA clinical observational study. (Refer to lines 247 to 250)

Caregivers whose child died in HDU did not participate in the surveys. A maximum of 110 caregivers were expected to be interviewed, dependent on the number of admissions and caregiver consent during the data collection period. (Refer to lines 256 to 258)

The sampling frame for the questionnaire-based healthcare provider survey comprised all healthcare providers (i.e., the complete census of nurses and clinicians) working in the HDU from January to May 2023. (Refer to lines 258 to 261)

Reviewer 1 comments: Consider discussing potential selection and courtesy bias in caregiver responses, especially since interviews were conducted post discharge.

Response: Thank you for bringing this point to our attention. We have added a paragraph describing how selection and interviewer bias were addressed in the section “Strengths and limitations of the study”, as follows:

We noted that interviewing caregivers after discharge may have introduced selection bias, as those with either more positive or more negative experiences could lead to an increased willingness to participate. However, all the invited caregivers consented to participate, such that selection bias remained limited. (Refer to lines 963 to 966)

Observer effects may also have influenced participants’ behaviour during the HDU observations. While researchers made efforts to minimise disruption and blend into the clinical environment, their presence may still have affected how staff interacted with the monitors. Additionally, caregiver responses may have been influenced by courtesy bias, particularly if gratitude towards healthcare providers shaped their feedback. To mitigate this, participants were assured at the start of each interview that their responses would remain anonymous and would not affect their future access to care. (Refer to lines 968 to 974.)

Reviewer 1 comments: Clarity and language: thorough language review is recommended to correct grammatical inconsistencies, repetitive phrasing, and tense shifts.

Response: Thank you for your comment. This has been addressed by reading and re-reading the manuscript, and making corrections accordingly.

Reviewer 1 comments: Please revise long or redundant sentences, particularly in the discussion section, to improve readability.

Response: Thank you for bringing this important point to our attention. The discussion has been double-checked for long and redundant formulations, and revised accordingly. (Refer to lines 793 to 941).

Reviewer 1 comment: Ethical procedure: Kindly specify who conducted the informed consent process and how it was administered, especially for illiterate participants.

Response: Thank you for your suggestion; we have included statements in the section “ethical consideration” to address the issue raised, as follows (Refer to lines 361 to 368)

Written informed consent, translated into the local language, was provided to all participants by DM, including caregivers of critically ill children and healthcare providers, for observations, in-depth interviews, and questionnaire-based surveys, with a signature or thumbprint. In cases of illiterate participants, DM read the information sheet and written consent form aloud in their local language. These participants were then asked to confirm their understanding, provide verbal consent voluntarily, and provide a thumbprint.

Reviewer 1 comment: Data Availability Statement: The current statement indicates that data are available upon request due to confidentiality. In line with PLOS ONE's open data policy, I recommend that the authors de-identify and publicly share applicable quantitative datasets or provide a clear justification for the restriction.

Response: The quantitative data set has been de-identified and will be publicly shared per the journals’ requirements.

Reviewer 1 comment: Statistical analysis: Where appropriate, please consider including confidence intervals or p-values to give more context to the descriptive statistics.

Response: Thank you for your suggestion. We have taken it on board and added confidence intervals to Tables 1 and 2, adjusting the text accordingly. (Refer to lines 452-465 for the discussion of table 1 results and to lines 518-525 for table 2 results).

Reviewers 1 comment: Clarify whether any comparative analyses were performed (e.g., before vs. after implementation, or between trained vs. untrained staff), even if exploratory.

Response: Thank you for your valuable comment. We have carefully considered this and included it in the section “Resu

---

## [Editor Report · Decision Letter 1]

15 Jul 2025

Implementing the IMPALA Continuous Monitoring System for Paediatric Critical Care in Malawi: A mixed methods study of barriers and facilitators

PONE-D-25-13594R1

Dear Dr. Mwale,

We’re pleased to inform you that your manuscript has been judged scientifically suitable for publication and will be formally accepted for publication once it meets all outstanding technical requirements.

Kind regards,

Nik Hisamuddin Nik Ab. Rahman

Academic Editor

PLOS ONE